# Spatial Risk Factors of Vector-Borne Diseases in Pacific Island Countries and Territories: A Scoping Review

**DOI:** 10.3390/tropicalmed11010006

**Published:** 2025-12-24

**Authors:** Tathiana Nuñez Murillo, Angela Cadavid Restrepo, Helen J. Mayfield, Colleen L. Lau, Benn Sartorius, Behzad Kiani

**Affiliations:** 1School of Public Health, Faculty of Health, Medicine and Behavioural Sciences, The University of Queensland, Brisbane, QLD 4006, Australia; a.cadavidrestrepo@uq.edu.au; 2UQ Centre for Clinical Research, Faculty of Health, Medicine and Behavioural Sciences, The University of Queensland, Brisbane, QLD 4029, Australia; h.mayfield@uq.edu.au (H.J.M.); colleen.lau@uq.edu.au (C.L.L.); b.sartorius@uq.edu.au (B.S.); b.kiani@uq.edu.au (B.K.)

**Keywords:** vector-borne diseases, arbovirus, climate change, environmental risk factors, Pacific Island Countries and Territories (PICTs), sociodemographic determinants, climatic factors, infectious disease

## Abstract

This scoping review aimed to identify and synthesise spatially relevant environmental, demographic, and socio-economic factors associated with vector-borne diseases (VBDs) in Pacific Island Countries and Territories (PICTs), a region particularly vulnerable due to its ecological and climate diversity. A systematic search of PubMed, Scopus, and Web of Science was conducted in March 2025 with no time restrictions, yielding 3008 records. After applying the inclusion criteria, 21 studies were selected for analysis. Environmental factors such as temperature, precipitation, and land cover were consistently associated with increased burden of malaria, dengue, and lymphatic filariasis, while associations with elevation and flooding were mixed or inconclusive. Demographic factors, including population density and household composition, were found to be associated with disease occurrence, although the direction and the strength of these associations varied. Three studies reported a negative association between population density and disease outcomes, including lymphatic filariasis in American Samoa and dengue in New Caledonia. Spatial socioeconomic factors such as low income, unemployment, and limited education were positively correlated with disease burden, particularly lymphatic filariasis and dengue. These findings underscore the importance of spatial determinants in shaping VBD transmission across PICTs and highlight the utility of spatial risk mapping to inform geographically targeted vector control strategies. Notably, infrastructure, health care access, and intra-island mobility remain underexplored in the literature, representing critical gaps for future research. Strengthening surveillance through spatially informed public health planning is essential to mitigate disease burden in this climate-sensitive and geographically dispersed region.

## 1. Introduction

The Pacific Island Countries and Territories (PICTs) are a group of small island landmasses dispersed across the vast Pacific Ocean, which covers nearly one-third of the Earth’s surface [1]. Despite their wide geographic distribution, the PICTs are home to approximately 11.4 million people and are predominantly tropical and subtropical in climate [2,3,4]. This vast expanse includes some of the most remote and least economically developed nations globally [3]. PICTs experience a disproportionate burden of climate change impacts due to the combined effects of high geographic exposure (e.g., low-lying coastal settlements), economic constraints, and limited adaptive capacity. These challenges manifest through rising sea levels, extreme weather events, and climate-sensitive livelihoods, and are further compounded by demographic and socioeconomic factors [5]. Collectively, these conditions underscore the urgent need for comprehensive risk assessments that prioritise the health and well-being of local populations, particularly given their amplified vulnerability to infectious diseases in the face of escalating environmental challenges [6].

Vector-borne diseases (VBDs) such as dengue, malaria, chikungunya, and lymphatic filariasis continue to pose significant public health challenges across PICTs [7]. On a global scale, VBDs account for approximately 17% of these communicable diseases and cause around 1 million deaths annually [7,8]. In the PICTs, the intersection of environmental vulnerability, human population dynamics, socio-economic disadvantage, and fragile infrastructure creates ideal conditions for VBD transmission [9,10]. The region experiences recurrent outbreaks driven by a complex interplay of these factors, with evolving climate conditions playing a central role. The burden of VBDs is multifaceted: epidemiologically, they contribute to high morbidity and add strain to already limited healthcare systems; environmentally, climate change intensifies vector proliferation and transmission cycles [11]; demographically, rapid human population growth and urbanisation in developing countries often lead to poor living conditions, including slums and informal settlements, which create densely populated environments that facilitate mosquito breeding and disease spread; and socioeconomically, high levels of poverty and limited health literacy further exacerbate vulnerability [12].

Spatial risk factors refer to the environmental, demographic, and socio-economic characteristics that vary across geographic space and influence where, how, and to what extent VBDs occur [13]. These include climatic variables such as high temperatures, rainfall, extreme weather events, and vegetation cover, as well as population-based factors like housing density, income, education levels, and healthcare access [14,15,16]. Together, these determinants compound population vulnerability by increasing exposure to vectors, limiting access to preventive measures, and delaying diagnosis and treatment [10,17]. Understanding the spatial distribution of such factors is critical for identifying high-risk areas, guiding resource allocation, and designing geographically tailored interventions. This is particularly important in the Pacific, where vast geographic dispersion, ecological diversity, and limited public health capacity present unique challenges [18]. 

Despite growing interest in the spatial epidemiology [19] of VBDs in the Pacific, the existing evidence base remains fragmented. Studies have investigated a wide range of environmental and sociodemographic drivers, yet their focus, methodology, and geographic coverage vary considerably. For instance, Park et al. (2016) in Papua New Guinea reported a positive association between malaria incidence and rainfall, but no association with elevation [20], whereas Descloux et al. (2012) demonstrated that increased relative humidity and temperature were linked to dengue incidence in New Caledonia [21]. Large-scale climate phenomena such as El Niño Southern Oscillation (ENSO) have also been implicated in driving outbreaks, as evidenced in studies by Hales et al. (1996) and Smith et al. (2017) [22,23]. Socio-demographic factors, though less frequently studied, have revealed important insights: Zellweger et al. (2017) found an inverse relationship between human population density and dengue incidence [24], suggesting that household-level infrastructure and housing quality may mitigate transmission risk. Similarly, Teurlai et al. (2015), in a study in New Caledonia, identified unemployment and low education levels as significant drivers of dengue incidence at the commune level [25].

Given this heterogeneity, a consolidated understanding of the spatial risk factors for VBDs in PICTs is urgently needed to inform geographically targeted and context-sensitive control strategies. This scoping review aims to synthesise the current body of literature examining environmental and sociodemographic factors for VBDs across the Pacific region. Specifically, this review synthesises the key variables assessed, describes relationships between contextual factors and disease outcomes, and highlights gaps in the evidence base to inform future research and policy. By providing a comprehensive overview of spatial drivers of VBD transmission and burden, this review offers insights essential for strengthening surveillance, prevention, and public health preparedness across PICTs. 

## 2. Methods

This scoping review is reported in accordance with the Preferred Reporting Items for Systematic Reviews and Meta-Analyses extension for Scoping Reviews (PRISMA-ScR) Checklist (Appendix A).

### 2.1. Protocol and Registration

The scoping review’s protocol was registered with the International Platform of Registered Systematic Review and Meta-analysis Protocols (INPLASY) in September 2025 (INPLASY 202590090).

### 2.2. Eligibility Criteria

Studies were included based on the following criteria: (a) original research, (b) inclusion of at least one spatially relevant factor (environmental, demographic, or socioeconomic), (c) include any VBDs, (d) studies conducted in the PICTs, (e) present quantitative statistical results, and (f) studies conducted in humans. No restrictions were applied regarding publication year.

### 2.3. Information Sources

The literature search was conducted in March 2025 across PubMed, Scopus, and Web of Science. All identified articles were imported into Covidence for study selection. Screening was performed at the title, abstract, and full-text levels. Discrepancies in study inclusion or data extraction were resolved through discussion and consensus among all authors to ensure consistency and methodological rigour. 

### 2.4. Search Strategy

The search combined terms related to VBDs (e.g., malaria, dengue, lymphatic filariasis), environmental, demographic, and socioeconomic factors (e.g., temperature, rainfall, population density, income), and geographic terms for PICTs. Searches were restricted to title and abstract fields, with no year limits applied. The full electronic strategies for PubMed, Scopus, and Web of Science are provided in Appendix A.

### 2.5. Data Extraction Process

For each article, the following data were extracted manually: (a) title, (b) citation details, (c) year of publication, (d) aim, (e) study design, (f) geographic scope, (g) study location, (h) disease studied, (i) outcome, (j) data sources, (k) environmental, demographic or socioeconomic factor, (l) key results, (m) statistical and modelling approaches, and (n) main conclusions. The extraction aimed to capture relevant information to enable the thematic synthesis. Data were recorded in a structured spreadsheet to ensure consistency and facilitate subsequent analysis. Discrepancies or ambiguities in the data extraction process were resolved through consensus among the authors. 

### 2.6. Data Synthesis and Analysis

Following data extraction, data were synthesised using a structured tabulation approach to identify relationships across environmental, demographic, and socioeconomic factors influencing disease burden in PICTs. Each study was categorised by factor type, nature of the relationship (positive or negative association), and disease studied. Where available, we also noted whether the associations were derived from bivariate (unadjusted) or multivariable-adjusted analyses (i.e., models with a single outcome and multiple covariates). The reporting of significance varied across studies, with results expressed through different metrics such as *p*-values, odds ratios, and credible intervals. By applying this structured tabulation method, we were able to generate a comparative overview of the evidence, consistent with scoping review methodology, and provide a comprehensive landscape of current research across the region. 

## 3. Results

The initial database search identified 3008 records. Covidence automatically removed duplicates, and three additional duplicates were identified and removed manually. After deduplication, 1933 unique records remained for screening. Of these, 1850 were excluded based on title and abstract screening. The full texts of the remaining 83 articles were retrieved and assessed for eligibility. Following the full text review, 62 articles were excluded due to one or more of the following reasons: absence of quantitative results, not being original research, lack of assessment of any spatial factors, inaccessibility of full text, absence of outcomes related to VBDs, or focus on non-human subjects or settings outside the Pacific Island region. A total of 21 studies met the inclusion criteria and were included in the review (Figure 1 and Appendix A). The included studies were published between 1998 and 2025. Geographically, studies were unevenly distributed across PICTs, with New Caledonia, Papua New Guinea, and American Samoa being the most frequently studied settings, indicating a strong geographic concentration of available evidence. Dengue and malaria were the most investigated diseases, followed by lymphatic filariasis. Ecological designs mixed with time-series and cross-sectional approaches predominated, and twelve studies assessed environmental correlations with lagged association. This heterogeneity in study design has important implications for interpretation: time-series and lagged ecological studies provide stronger insight into climate–disease dynamics, whereas cross-sectional studies primarily support spatial correlation rather than causal inference. Data sources varied widely, including national health surveillance systems, meteorological records, and survey data, highlighting the diversity of methodological approaches used to assess spatial risk factors, but also contributing to variability in the strength and comparability of reported associations across settings.

Among the environmental factors, temperature was the most frequently examined factor, with ten studies (47.62%), followed by 6 studies (28.6%) on precipitation, assessing its influence on VBD burden (Table 1). Most studies reported a positive association between temperature metrics and VBD incidence, particularly for malaria and dengue, with the magnitude of effects ranging from threshold-based outbreak triggering to incremental increases in incidence rates. Ochida et al. (2022) [26] identified a non-linear relationship between dengue outbreak risk and both temperature and precipitation, with temperature showing a stronger and more monotonic effect. In their model, epidemic weeks were defined by an effective reproduction number RT, greater than 1, and the probability of RT > 1 increased markedly when the number of days with maximum temperature exceeded 30.8 °C during the preceding 80 days [26], indicating a temperature threshold above which sustained transmission becomes more likely. Similarly, Nelson et al. (2022) [4] found a positive association between suspected dengue cases and climatic variables in Fiji using Poisson generalised linear models (GLM). Temperature explained 23.8–38.2% of the variation, although it was not statistically significant [4]. In contrast, Riou et al. (2017) found no effect of temperature on transmission of Chikungunya and Zika in French Polynesia and the French West Indies; instead, they reported a dual delayed effect of precipitation, first decreasing and then increasing transmission risk [27]. Chaves et al. (2008) reported a significant positive association between temperature and malaria incidence in Vanuatu before the widespread use of insecticide-treated nets, with each 1 °C increase (at 3-month lag) raising the incidence rate of *Plasmodium falciparum* by approximately 1.43 cases per 1000 population (*p* < 0.05) [28], representing a clinically meaningful effect size at the population level in an endemic island setting.

In general, the evidence indicates that temperature might influence VBD transmission both through non-linear outbreak thresholds and through steady increases in endemic incidence, with the strength and statistical significance of associations varying by disease, geographic setting, and modelling approach (e.g., GLM, Support Vector Machine (SVM), time-series, and mechanistic models).

Rainfall and precipitation were also key variables, and each was examined in eleven studies (Table 1). These factors consistently demonstrated a positive correlation with disease incidence, especially for malaria, dengue, chikungunya, and Zika, with effect sizes ranging from modest incremental increases in regression-based models to stronger delayed effects in time-series and Bayesian frameworks. Notably, studies such as Andhikaputra et al. 2023 and Nelson et al. 2022 highlighted the role of lag periods in modulating the timing and strength of the relationship between rainfall and disease transmission for dengue-like illness and suspected dengue; while Andhikaputra et al. reported a positive and significant association at lag 0 (i.e., same-week exposure) across all islands, Nelson observed the strongest positive effect after a 5-week lag, indicating a delayed climatic influence on transmission [4,29], consistent with the biological delay required for vector breeding and viral incubation following rainfall events. Similarly, Park et al. (2016) observed concurrent increases in malaria incidence and rainfall over 1996–2008 in Eastern Highlands Province, Papua New Guinea, with rainfall trending upwards and positively associated with incidence [20], while Imai et al. (2016) found a contrasting pattern in the same region, where precipitation exhibited a significant negative association with malaria incidence at lags of 0, 1, and 2 months (*p* < 0.05) [30], suggesting that in high-altitude settings heavy rainfall may produce a flushing effect that disrupts larval habitats. In addition, Ochida et al. (2022) reported a positive but non-linear association between precipitation and dengue incidence [26], indicating that transmission risk increases only beyond specific rainfall thresholds rather than in a strictly proportional manner. Furthermore, Bayesian modelling in Sorenson et al. (2025) in Vanuatu showed that at a two-month lag, total precipitation had a positive and statistically significant effect on malaria incidence (mean coefficient = 0.19, 94% HDI: 0.11–0.28) [31], confirming a positive delayed association between precipitation and malaria incidence in this setting.

The measure of climatic patterns using the Southern Oscillation Index (SOI) and ENSO events, assessed in only two studies, was positively associated with dengue incidence and negatively associated with malaria disease incidence [22,23], potentially indicating that large-scale climate variability may influence different vector-pathogen systems in opposite directions depending on local ecological conditions. In contrast, flooding, examined in a single study conducted in Fiji, showed no clear association with disease outcomes, specifically seroprevalence and incidence of dengue, chikungunya, and Zika [32], suggesting that short-term extreme rainfall events alone may be insufficient to modify transmission risk in the absence of sustained changes in vector habitat or exposure patterns, particularly when measured using cross-sectional study designs.

Land cover characteristics, including urbanisation and vegetation density, were associated with increased risk for diseases such as lymphatic filariasis in American Samoa, reported in two studies, and dengue, reported in one study. Tree coverage showed the strongest effect, with OR 1.00 (95% CrI 1.001–1.01) from the Bayesian model, indicating that each unit increase in tree coverage was associated with a corresponding increase in the odds of disease occurrence compared with areas of lower vegetation coverage. Similarly, the Poisson model yielded a relative risk of 1.06 (95% CI 1.05–1.08, *p* < 0.01), reflecting a 6% increase in incidence per unit increase in vegetation density. Together, these estimates highlight the role of both natural and built environments [15,16,24], with effect sizes representing small but consistent increases in risk per incremental increase in vegetation coverage that may translate into substantial changes in transmission intensity at the population level.

Elevation showed inconsistent effects; higher altitudes were generally associated with lower prevalence of diseases such as malaria and lymphatic filariasis [16,20,33,34], consistent with the known influence of elevation as a proxy for lower temperatures and reduced vector survival and breeding suitability. However, a study conducted by Cadavid Restrepo et al. (2023) in American Samoa [15] reported no significant association, suggesting that in relatively low-elevation island settings with limited altitudinal gradients, elevation may not exert a strong modulatory effect on transmission risk.

Spatial demographic factors, such as population density measured at the community level in three studies, and human density measured as people per household in two studies (Table 2), also play a critical role and have shown varied associations with disease outcomes across PICTs, highlighting the importance of the spatial scale at which population structure is measured. Studies consistently reported a negative relationship between population density and disease outcomes, including lymphatic filariasis prevalence in American Samoa and dengue incidence in New Caledonia [16]. For lymphatic filariasis in American Samoa, population density was negatively associated with seroprevalence (RR = 0.98, 95% CI: 0.96–0.99) in a multivariable Poisson regression model [16] and showed a non-significant negative trend for prevalence in a Bayesian geostatistical model (OR = 0.96, 95% CrI: 0.86–1.06) [15]. Similarly, dengue incidence in New Caledonia demonstrated a negative association with population density (IRR = 0.88, 95% CI: 0.77–1.00) in univariable analysis [24]. Human density has shown mixed associations with dengue incidence; while one study found a strong positive correlation (r = 0.739, *p* < 0.0001) for people per household [25], another reported a negative one [24], both conducted in New Caledonia, suggesting that household-level crowding may amplify transmission risk even in settings where broader area-level population density appears protective.

Being born in the Pacific was positively associated with higher dengue incidence, which might reflect differences in lifestyle and habits [24]; moreover, people might spend more time outdoors, increasing their exposure, indicating that behavioural and cultural factors may modify demographic risk patterns. Finally, genetic factors such as the G6PD deficiency rate aggregated at the island level have been linked to increased malaria incidence and prevalence in Vanuatu [35], illustrating how population-level genetic susceptibility can interact with environmental exposure to shape endemic transmission patterns.

The influence of socioeconomic factors on VBD transmission in PICTs reveals a dynamic interaction of economic vulnerability (Table 3) that can intensify the impact of broader social conditions, with effect sizes indicating a meaningful contribution to population-level risk. Interestingly, income level, assessed in one study, showed that lower income was associated with higher lymphatic filariasis seroprevalence in American Samoa, with an adjusted odds ratio of 1.46 (95% CI: 1.02–2.11) [36], representing a moderate independent effect after adjustment for other covariates. Lower employment status and educational attainment were also consistently linked to higher dengue incidence; unemployment was associated with higher disease burden [24,25], suggesting that limited economic resources may constrain access to preventive measures and timely healthcare. However, findings regarding education level are mixed. While Zellweger et al. (2017) found a positive association in lower education levels and dengue incidence, Teurlai et al. (2015) found no significant relationship [24,25], indicating that the influence of education may be context-specific and sensitive to how socioeconomic indicators are measured and modelled across studies.

**Table 1 tropicalmed-11-00006-t001:** Environmental Factors Associated with Vector-Borne Disease Outcomes in Pacific Island Countries and Territories (PICTs).

Environmental Factors *n* = 34	Disease	Measurement		Study Design/ Location	Statistical and Modelling Approaches	Association	Outcome	Study Reference
Temperature	Dengue	Daily min/max, mean	1–2 month lag	Ecological/ New Caledonia	Spearman’s rank correlation and multivariable non-linear support vector machine (SVM) model	Positive	Incidence rate	Descloux et al. 2012 [21]
	Dengue	Daily maximum	3-month lag	Ecological/ New Caledonia	Effective reproduction number + SVM	Positive NL ^1^	Outbreak	Ochida et al. 2022 [26]
	Suspected dengue	Weekly mean Min/max	No mention	Ecological/ Fiji	Generalised Linear Model (GLM)	Positive	Incidence	Nelson et al. 2022 [4]
	Chikungunya, Zika *	Weekly mean	1-month lag	Retrospective comparative modelling/ French Polynesia	Time-Series Susceptible–Infected–Recovered model	Positive NS ^2^	Outbreak	Riou et al. 2017 [27]
	Dengue	Monthly mean	No mention	Ecological/ New Caledonia	Pearson correlation + SVM	Positive	Incidence	Teurlai et al. 2015 [25]
	Malaria	Monthly min/max	No mention	Retrospective ecological time series/ Papua New Guinea	Linear regression	Positive	Incidence	Park et al. 2016 [20]
	Dengue-like illness	Monthly mean	1-month lag	Cross sectional/ Solomon Islands	Negative binomial regression, Generalised Estimating Equations	Positive	Incidence	Andhikaputra et al. 2023 [29]
	Malaria	Monthly Min/max mean	1–2-month lag	Retrospective ecological time-series/ Vanuatu	GLM and Bayesian model	Positive	Incidence	Sorenson et al. 2025 [31]
	Malaria	Monthly Minimum	3-month lag	Cross-sectional/ Papua New Guinea	GLM	Positive	Incidence	Imai et al. 2016 [30]
	Malaria	Monthly mean	3-month lag	Ecological/ Vanuatu	Spatial Autoregressive Model	Positive	Incidence rate	Chaves et al. 2008 [28]
Rainfall	Malaria	Mean daily (mm)	No mention	Ecological/ New Caledonia	Pearson correlation + SVM	Positive	Incidence	Teurlai et al. 2015 [25]
	Suspected dengue	Mean weekly rainfall (mm)	1–2-month lag	Ecological/ Fiji	GLM	Positive	Incidence	Nelson et al. 2022 [4]
	Malaria	Monthly total rainfall (mm)	No mention	Retrospective ecological time-series/ Solomon Islands	Linear regression + bootstrap	Positive	Incidence	Smith et al. 2017 [23]
	Dengue-like illness	Monthly Cumulative rainfall (mm)	1-month lag	Cross-sectional/ Solomon Islands	Negative binomial regression + Generalised Estimating Equations	Positive	Incidence	Andhikaputra et al. 2023 [29]
	Malaria	Annual total rainfall (mm)	No mention	Retrospective ecological time series/ Papua New Guinea	Linear regression	Positive	Incidence	Park et al. 2016 [20]
Precipitation	Dengue	Mean daily (mm/day)	2–3-month lag	Retrospective ecological modelling/ New Caledonia	Effective reproduction number + SVM	Positive NL ^3^	Outbreak	Ochida et al. 2022 [26]
	Chikungunya, Zika	Mean weekly (cm)	1–2-week/5-week lag	Retrospective comparative modelling/French Polynesia	Time-Series Susceptible–Infected–Recovered model	Negative/ positive	Outbreak	Riou et al. 2017 [27]
	Malaria	Mean monthly (mm/month)	0–2-month lag	Cross-sectional/ Papua New Guinea (Madang)	GLM	Positive	Incidence	Imai et al. 2016 [30]
	Zika	Mean monthly (mm/month)	No mention	Retrospective mathematical modelling/ French Polynesia	Mathematical transmission model	Negative	Outbreak	He et al. 2017 [37]
	Malaria (*P. falciparum*)	Mean monthly (mm/month)		Cross-sectional/Papua New Guinea	GLM and Bayesian Decision Network (BDN) Modelling	Positive	Prevalence	Cleary et al. 2021 [33]
	Malaria	Total monthly (mm/day)	1-month lag	Retrospective ecological time-series/ Vanuatu	GLM	Negative NS	Incidence	Sorenson et al. 2025 [31]
			2-month lag		Bayesian model	Positive		
Humidity	Dengue	Maximal relative humidity (%)		Ecological/ New Caledonia	Spearman’s rank correlation + multivariable non-linear SVM model	Positive	Incidence rate	Descloux et al. 2012 [21]
Southern Oscillation Index (SOI)	Dengue	El Nino Southern Oscillation (ENSO)		Mixed ecological/ 14 South Pacific Islands	Pearson correlation	Positive	Incidence	Hales et al. 1999 [38]
	Malaria			Retrospective ecological time-series/ Solomon Islands	Linear regression + bootstrap	Positive	Incidence	Smith et al. 2017 [23]
Flooding	Dengue, Chikungunya, Zika	Flooding		Cross-sectional/ Fiji	Logistic regression	Negative NS ^4^	Seroprevalence/Incidence	Rosser et al. 2025 [32]
Landcover	Lymphatic filariasis	Cropland		Cross-sectional/ American Samoa	Bayesian geostatistical logistic regression	Positive NS ^5^	Prevalence	Cadavid Restrepo et al. 2023 [15]
		Built/Urban				Positive NS ^6^		
		Tree coverage				Positive NS ^7^		
	Lymphatic filariasis	Built/Urban		Cross-sectional/ American Samoa	Multivariable Poisson regression model	Strong Positive	Seroprevalence	Lemin et al. 2022 [16]
		Tree coverage						
		Rangeland						
	Dengue	Vegetation coverage		Ecological/ New Caledonia	Univariable regression analysis	Positive	Incidence	Zellweger et al. 2017 [24]
Elevation	Malaria	>1700 m (altitude)		Retrospective ecological time- series/ Papua New Guinea	Linear regression	Negative NS ^8^	Incidence	Park et al. 2016 [20]
		1500–1699 m				Positive		
	Lymphatic filariasis	Mean 77.72 (m)		Cross-sectional/ American Samoa	Bayesian geostatistical logistic regression	Positive NS ^9^	Prevalence	Cadavid Restrepo et al. 2023 [15]
	Malaria (*P. vivax*)	Above sea level		Cross-sectional/ Papua New Guinea	GLM	Negative	Prevalence	Cleary et al. 2021 [33]
	Malaria	Per 10 m		Cross-sectional/ Papua New Guinea	Multivariable logistic regression	Negative	Prevalence	Myers et al. 2009 [34]
	Lymphatic filariasis	Slope gradient (degrees)		Cross-sectional/ American Samoa	Multivariable Poisson regression model	Negative	Seroprevalence	Lemin et al. 2022 [16]

*n*: The total number of studies included in the literature review that analysed the variable; ^1^ Non-linear (NL), ^2^ No significant (NS), Confidence intervals (CI): * Chikungunya: (0.30–0.55) Zika: (0.13–0.30), Leave-One-Out Information (LOOIC), ^3^ Non-linear: (NL), ^4^ Non-significant (NS), Adjusted Odds ratio (aOR): 0.5, Confidence interval (CI): (0.2–1.4), ^5^ No significant (NS): Odds ratio (OR): 1.06: Credible interval (CrI): (0.97–1.15), ^6^ NS OR: 1 CrI: (0.99–1.00), ^7^ NS OR: 1 CrI: (1.001–1.01), ^8^ NS *p*-value: 0.2894, ^9^ NS OR: 1 CrI: (0.99–1.00).

**Table 2 tropicalmed-11-00006-t002:** Demographic Factors Associated with Vector-Borne Disease Outcomes in Pacific Island Countries and Territories (PICTs).

Demographic Factors *n* = 7	Disease	Measurement	Study Design/ Location	Statistical Measure	Association	Outcome	Study Reference
Population density	Lymphatic Filariasis	persons/km^2^	Cross-sectional/ American Samoa	Multivariable Poisson regression model	Negative	Seroprevalence	Lemin et al. 2023 [16]
	Lymphatic Filariasis	people/m^2^	Cross-sectional/ American Samoa	Bayesian geostatistical logistic regression	Negative	Prevalence	Cadavid Restrepo et al. 2023 [15]
	Dengue	N° of inhabitants/km^2^	Ecological/ New Caledonia	Univariable regression analysis	Negative	Incidence	Zellweger et al. 2017 [24]
Human density	Dengue	People per household	Ecological/ New Caledonia	Pearson correlation + SVM ^1^	Strongly positive	Incidence	Teurlai et al., 2015 [25]
	Dengue		Ecological/ New Caledonia	Univariable regression analysis	Negative	Incidence	Zellweger et al. 2017 [24]
Born in the Pacific	Dengue	Not applicable	Ecological/ New Caledonia	Univariable regression analysis	Positive	Incidence	Zellweger et al. 2017 [24]
Other factors	Malaria	(G6PD) ^2^	Observational/ Vanuatu	Spearman’s rank Correlation	Positive	Incidence/Prevalence	Kaneko et al. 1998 [35]

^1^ Support Vector Machine; ^2^ (G6PD) Glucose 6-phosphate dehydrogenase deficiency.

**Table 3 tropicalmed-11-00006-t003:** Socioeconomic Factors Associated with Vector-Borne Disease Outcomes in Pacific Island Countries and Territories (PICTs).

Socioeconomic Factors *n* = 6	Disease	Measurement	Study Design/ Location	Statistical Measure	Association	Outcome	Study Reference
Income	Lymphatic Filariasis	Low income	Ecological/ American Samoa	Multivariable logistic regression	Positive	Seroprevalence	Graves et al. 2020 [36]
Employment	Dengue	Unemployment	Ecological/ New Caledonia	Pearson correlation + SVM ^1^	Positive	Incidence	Teurlai et al. 2015 [25]
	Dengue		Ecological/ New Caledonia	Univariable regression analysis	Positive	Incidence	Zellweger et al. 2017 [24]
Educational level	Dengue	low education	Ecological/ New Caledonia	Univariable regression analysis	Positive	Incidence	Zellweger et al. 2017 [24]
	Dengue	No mention	Ecological/ New Caledonia	Pearson correlation + SVM	Positive NS *	Incidence	Teurlai et al. 2015 [25]
Infrastructure	Dengue	Old buildings/cement lodgings	Ecological/ New Caledonia	Univariable regression analysis	Positive	Incidence	Zellweger et al. 2017 [24]

* Non-significant (NS) *p*-value: 0.4529; ^1^ Support Vector Machine.

## 4. Discussion

This review synthesised evidence on the determinants of VBD transmission across PICTs, highlighting the complex and multivariable interactions among environmental, sociodemographic, and household factors. Within this broader framework, temperature, rainfall, and socioeconomic status consistently emerged as the most influential variables, with several studies demonstrating their association with VBD burden. These findings reinforce the need to investigate VBD epidemiology through an integrated perspective that accounts for the combined influences of climatic and social determinants. Additionally, these findings underscore the potential value of incorporating spatially resolved environmental and sociodemographic data into surveillance systems to improve outbreak prediction and guide targeted interventions. From a public health decision-making perspective, this supports the use of spatial risk profiling to prioritise high-risk communities for vector control, seasonal preparedness planning, and targeted health promotion, particularly in resource-constrained island settings.

Environmental factors were consistently highlighted across the reviewed studies, with temperature and precipitation most frequently examined. Temperature affects vector biology by accelerating mosquito development, increasing biting rates, and shortening pathogen incubation within vectors [5]. In general, higher temperatures were positively associated with disease incidence, particularly for malaria and dengue, and these effects were often shaped by seasonal patterns, as warmer and wetter periods tend to enhance vector activity. Nevertheless, exceptions were noted; for example, Riou et al. (2017) found that temperature alone was not a reliable predictor across all contexts [27]. Descloux et al. (2012) [21] have shown that the relationship between climatic variables and dengue is not strictly linear. Outbreak risk increases under specific configurations, such as sustained maximum temperatures >32 °C combined with low humidity, or moderate temperatures with very high humidity. This suggests that the interaction between climatic factors is more relevant than each variable in isolation [21]. Evidence from Vanuatu suggests that the influence of temperature on malaria transmission can be substantially reduced following effective vector control interventions [28], highlighting the role of public health measures in increasing system resilience to climatic variability. These findings indicate that temperature should not be considered in isolation, nor assumed to exert a simple linear effect on disease burden. Instead, it should be analysed in conjunction with other climatic variables to more accurately capture the complexity of disease dynamics. 

Precipitation is also critical due to the creation of breeding sites for vectors, particularly mosquitoes [11]. Standing water resulting from rain provides ideal conditions for mosquito larvae to develop [11], facilitating the transmission of VBDs; the consistent positive correlation found across the included studies supports this mechanism, indicating that increased standing water may enhance vector breeding and survival. Nelson et al. (2022) found that rainfall, with a lag of several weeks, significantly predicted variation in water-related disease syndromes in Fiji, although the explained variance was modest (maximum ~7.6% for some conditions) [4]. However, Park et al. (2016) reported a positive association in Eastern Highland Province, but in the coastal regions of Papua New Guinea, decreased rainfall was associated with reduced malaria incidence [20]. These contrasting findings highlight the precipitation effects, with evidence from Imai et al. (2016) [30] suggesting that in high-altitude areas such as Eastern Highlands, heavy rainfall may lead to a flushing effect that reduces mosquito breeding sites, resulting in a negative association with malaria incidence. These findings underscore the importance of incorporating temporal aspects in future studies, especially those focused on highly seasonal diseases in other regions, and reinforcing rainfall as a predictor when combined with temperature in probabilistic models.

Furthermore, the use of climatic indices such as the Southern Oscillation Index (SOI) and El Niño Southern Oscillation (ENSO) reinforced the hypothesis that large-scale climate phenomena can act as drivers of VBDs outbreaks in the region. Notably, the direction of association varied: La Niña conditions (SOI-positive) were linked to increased dengue incidence [38], whereas El Niño conditions (SOI-negative) were associated with higher malaria transmission in Northern Guadalcanal [23]. These findings highlight the value of integrating such indices into climate-informed early warning systems to anticipate periods of elevated transmission risk and to support ministries of health in the timely implementation of pre-emptive vector control, surge staffing, and community awareness campaigns.

Mixed findings across studies, referring to inconsistent or directionally different effects of specific factors, may partly reflect differences in methodological approaches, including variation in the spatial scale of analysis, for instance at the neighbourhood compared to regional level. In addition, the measurement of climatic and environmental variables is often more objective and quantitative, typically derived from standardised instruments, whereas demographic and socioeconomic factors are more prone to heterogeneity due to differences in case definitions, scoring systems, and categorisation across studies. Beyond these methodological considerations, several underlying mechanisms may further explain variability in effects. For example, overcrowding may increase contact rates and facilitate vector–human interaction, while being born in the Pacific may reflect behavioural or cultural patterns that influence exposure, such as time spent outdoors [24]. Genetic predispositions, such as G6PD deficiency reported in 16 villages in Vanuatu, further illustrate how long-term endemic exposure may shape population-level vulnerability [35]. Together, these insights can inform more effective and context-specific public health interventions tailored to the unique characteristics of island populations.

Socioeconomic variables also demonstrated significant associations with VBD transmission. Income level and employment status were positively associated with disease burden, reinforcing the notion that lower socioeconomic status may limit access to healthcare, vector control, and education, which can contribute to increased exposure to VBDs [24,25,36]. These associations suggest that socioeconomic disadvantage not only affects exposure risk but may also influence health-seeking behaviour and the ability to engage in protective practices.

Although the transmission dynamics of VBDs are inherently complex and shaped by interactions among climatic, environmental, demographic, and socioeconomic factors, the present study was designed with the primary aim of mapping the breadth of spatially relevant quantitative evidence in the PICTs. Accordingly, our eligibility criteria required only the inclusion of at least one spatially relevant factor and the reporting of quantitative statistical associations. This inclusive approach allowed us to capture studies with varying levels of analytical sophistication, including both bivariate and multivariable analyses. Importantly, the observed heterogeneity and, in some cases, inconsistency in reported effects across studies should not be interpreted as methodological shortcomings warranting exclusion, but rather as a key synthesis finding, reflecting the limited consideration of confounding and interaction effects in parts of the current evidence base. These findings highlight an important methodological gap in the region and underscore the need for future analytical studies and systematic reviews to explicitly model interaction structures and confounding mechanisms to better reflect the known complexity of vector-borne disease systems.

Despite growing evidence on the environmental and social determinants of VBDs, critical gaps persist in the literature. Chief among these is the limited exploration of healthcare access as a determinant of disease risk [39,40], a dimension that remains strikingly under-explored. Importantly, healthcare access should not be interpreted solely as physical or geographic accessibility [41], but also in terms of the capacity, quality, and functional readiness of health systems, including workforce availability, continuity of services, surveillance infrastructure, and policy implementation [42]. In many PICTs, structural health system constraints may substantially limit not only access to prevention and treatment, but also the effectiveness and timeliness of care, particularly during climate-sensitive outbreaks and extreme weather events [6]. This represents an additional layer of disproportional vulnerability when compared with high-income Western health systems that typically have greater surge capacity and system resilience.

The limited consideration of healthcare access and system capacity is further exacerbated by broader methodological weaknesses in the literature, including inconsistencies in how spatial factors are defined and measured, the lack of harmonised statistical approaches, and a pronounced geographic bias, with most studies concentrated in Papua New Guinea, New Caledonia, and American Samoa. Collectively, these shortcomings constrain the comparability of findings and obscure the regional picture. Future research must therefore move beyond these limitations by integrating healthcare access alongside health system capacity and quality as central determinants, adopting standardised spatial and statistical methods, and broadening geographic coverage to generate a more comprehensive and equitable evidence base.

This scoping review has several notable strengths. A comprehensive search strategy was applied across three major databases without restriction, ensuring broad coverage of relevant literature. The use of Covidence further strengthened methodological rigour by supporting systematic study selection and structured data extraction, thereby reducing error and enhancing consistency. In addition, the inclusion of environmental, demographic, and socioeconomic variables enabled a multidimensional analysis of transmission dynamics, revealing the complex interplay of these determinants. While some studies could not be included due to inaccessible full texts, this primarily limited the scope of factors captured. Overall, this review provides a valuable foundation for advancing research on spatial drivers of vector-borne diseases in the region and highlights the need for continued efforts to expand the evidence base. 

## 5. Conclusions

This scoping review provided a comprehensive synthesis of spatial determinants that influence the transmission and burden of VBDs across PICTs. The results highlighted the complex interaction, shaped by environmental as well as demographic and socio-economic factors, including seasonal events, population density, income levels, and educational literacy. While environmental conditions facilitate vector proliferation, demographic and socioeconomic vulnerabilities exacerbate exposure and limit access to preventive measures. The review also identified critical gaps in the current literature, particularly the underrepresentation of healthcare access as a determinant of disease risk. Addressing these gaps is essential for strengthening VBD surveillance and control. Future efforts should prioritise the integration of spatial risk assessments into public health planning. Policymakers can leverage this evidence to design more equitable and specific interventions, ensuring that surveillance and response strategies are tailored to the unique ecological and social contexts of the Pacific region.

## Figures and Tables

**Figure 1 tropicalmed-11-00006-f001:**
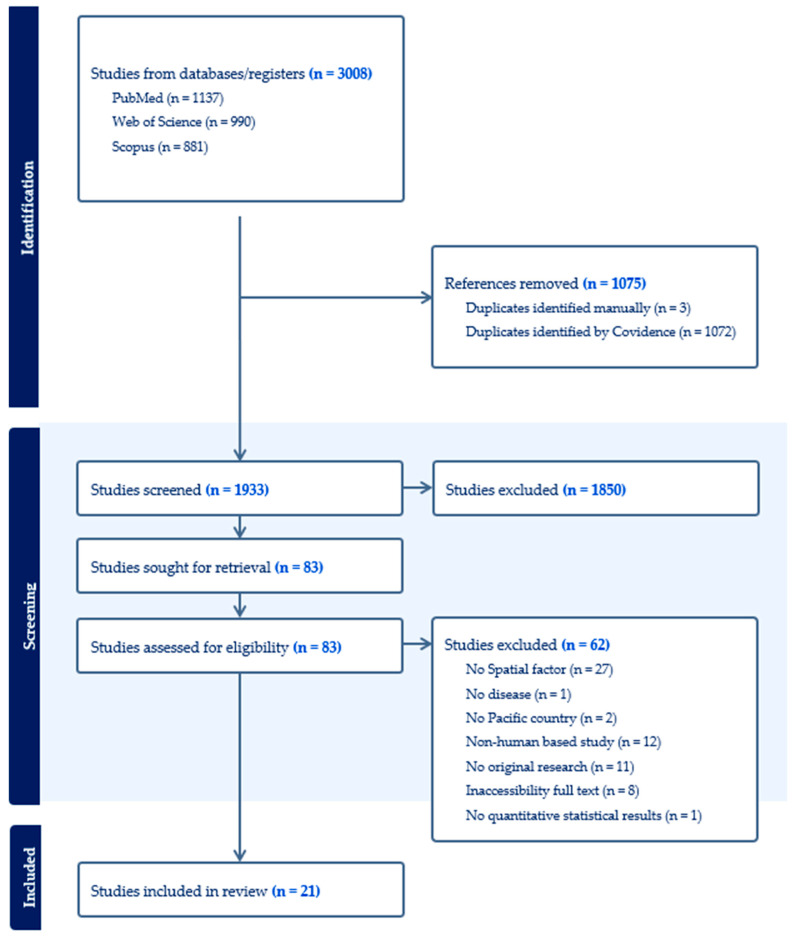
PRIMA Flow Diagram of the Literature Search and Screening Process.

## Data Availability

All data analysed during this scoping review are included in this published article and its Appendix A, including tables of the studies analysed, the extraction table, and the search strategy.

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
