# Peer review of "Spatial Risk Factors of Vector-Borne Diseases in Pacific Island Countries and Territories: A Scoping Review"

_tropicalmed, 2025, doi:10.3390/tropicalmed11010006_

Round 1

Reviewer 1 Report

Comments and Suggestions for Authors

The manuscript “Spatial Risk Factors of Vector-Borne Diseases in Pacific Island Countries & Territories: A Scoping Review” addresses a highly relevant topic and presents a broad, well-conceived introduction along with clearly defined methods. Accordingly, the study provides interesting insights that are consistent with the objectives of a scoping review.

However, the depth with which the findings are presented requires improvement in several aspects. Even in a scoping review, the numerical presentation of results should be more substantive. Merely stating whether associations were positive or negative is insufficient and does not reflect the full richness of the evidence reported by the included studies.

Moreover, the manuscript treats the included studies somewhat generically, without adequately exploring how differences in study design and geographical setting may influence the interpretation of findings. The synthesis should more explicitly acknowledge these differences and assess whether the consistency or strength of associations varies according to location and methodological approach.

Furthermore, although p-values and measures of association are listed in Table 1, they are scarcely explored in the narrative section. The same applies to non-linear associations, which are briefly mentioned but not adequately interpreted. It is therefore necessary to expand the interpretation of the statistical metrics already provided, highlighting the magnitude and consistency of associations across studies and discussing their contextual significance.

Finally, the discussion should be further developed after restructuring the results. It should also more clearly articulate the implications of the study’s findings and how they contribute to decision-making in public health.

Author Response

Reviewer's comment: The manuscript “Spatial Risk Factors of Vector-Borne Diseases in Pacific Island Countries & Territories: A Scoping Review” addresses a highly relevant topic and presents a broad, well-conceived introduction along with clearly defined methods. Accordingly, the study provides interesting insights that are consistent with the objectives of a scoping review.

However, the depth with which the findings are presented requires improvement in several aspects. Even in a scoping review, the numerical presentation of results should be more substantive. Merely stating whether associations were positive or negative is insufficient and does not reflect the full richness of the evidence reported by the included studies.

Moreover, the manuscript treats the included studies somewhat generically, without adequately exploring how differences in study design and geographical setting may influence the interpretation of findings. The synthesis should more explicitly acknowledge these differences and assess whether the consistency or strength of associations varies according to location and methodological approach.

Furthermore, although p-values and measures of association are listed in Table 1, they are scarcely explored in the narrative section. The same applies to non-linear associations, which are briefly mentioned but not adequately interpreted. It is therefore necessary to expand the interpretation of the statistical metrics already provided, highlighting the magnitude and consistency of associations across studies and discussing their contextual significance.

Authors’ response: We thank the reviewer for this valuable comment. In response, we have substantially strengthened both the tabular presentation and the narrative interpretation of the Results (highlighted in the manuscript).

First, we added two new columns (“Study design” and “Statistical and modelling approaches”) to all results tables, allowing readers to directly assess how analytical approaches influence reported associations.

Second, we revised the Results text (highlighted in the manuscript file) throughout to move beyond reporting only the direction of associations. We now explicitly interpret:

  • The magnitude of effects (e.g., small vs. moderate effect sizes),
  • Non-linear and lagged relationships (particularly for temperature and rainfall),
  • And how differences in study design and geographic setting influence consistency and interpretation of findings.

We also clarified scale-dependent effects (e.g., area-level population density vs. household crowding) and corrected the interpretation of key socioeconomic results (e.g., low income and lymphatic filariasis risk).

These revisions improve the depth, clarity, and interpretability of the Results, while remaining consistent with the scope of a scoping review.

Reviewer's comment: Finally, the discussion should be further developed after restructuring the results. It should also more clearly articulate the implications of the study’s findings and how they contribute to decision-making in public health.

Authors’ response: we have substantially revised and expanded the Discussion section (highlighted in the manuscript) following the restructuring of the Results. The Discussion now more clearly articulates:
(i) how the observed spatial, environmental, and socioeconomic patterns contribute to public health decision-making in Pacific Island Countries and Territories;
(ii) the implications for surveillance, climate-sensitive early warning systems, and targeted vector control strategies; and
(iii) the importance of accounting for context-specific, non-linear, and scale-dependent drivers when translating evidence into policy.

Reviewer 2 Report

Comments and Suggestions for Authors

Dear authors, 

The submitted manuscript provides a well-designed, properly executed and thoroughly, at least according to set objectives, review; charting up-to-date literature and comprehensibly summarizing relevant existing research-based evidence. Practically, it can be said that its primary purpose as a scoping review has been fulfilled to a large degree. 

Nevertheless, you will find below some general and specific comments and remarks, aiming to further enhance the validity and quality of your work. 

Line 39: For the shake of uniformity and consistency in the use and definition of PICT acronym, “countries” should be written as “Countries”. 

Lines 41-43: I believe a misinterpretation has taken place here. I believe, according to available information, this ratio is incorrect. The landmass area of PICTs is approximately 566.000 km2 within a broader ocean area of 29*106 km2. Actually, the Pacific Ocean’s surface coverage is one-third of the Earth’s surface (165.2 * 106 km2 / 510*106 km2, respectively). Please amend.  

Line 46: Disproportionately in what terms? I agree with your stated opinion, but it remains unclear what this disproportionate “ratio” refers to. Is it in terms of land surface coverage, population density, level of economic development or status, produced environmental impact of their socioeconomic activity etc.? I believe the imposed exposure and impact of climatic change is disproportionate in all of these senses and more, thus a more integrated and global approach will be needed to mitigate this burden. 

Lines 145-146: It is frequent in analytical studies, for most readers to perceive and several researchers to use the terms ‘multi-variate” and “multi-variable” interchangeably or as synonyms erroneously. In fact, these two terms refer to and define different situations; the former concerns statistical analyses procedures or models that encompass the simultaneous observation and analysis of more than one outcome variable (with potentially multiple independent variables), whereas the second term involves one outcome (or dependent) variable and multiple independent (predictor) variables. Judging from the content of Table 1 and 2, through the “Disease” and “Outcome” columns, I assume you refer to studies that investigated more than one VBD and/or measured more than one measure of disease as outcome, thus using the term as suggested above. If this is the case, and since both terms are mentioned in your manuscript (see also line 289), it would be useful to avoid the preservation and continuation of such misconceptions, by briefly explaining or defining the actual use of the term. The use of the term “multivariable” in line 289, seems appropriate and sensible though, in the provided context of the rationale of pose statement. 

Lines 301-309: You previously reported in the inclusion criteria of the selected and retained literature, the application of either bivariate or multivariate analyses along with the identification or verification of complex and multivariable interactions among the various types of potential risk factors, through synthesized evidence of the revised literature. Yet, some exceptions from consistent effects of these factors were observed. According to your statement (lines 306-309) such sources are implied to have implemented methodology (either in data collection and/or analysis) that cannot account properly or adequately the interaction or confounding effects of relevant factors (i.e. as temperature with other climatic factors. Shouldn't this be considered a priory as an inclusion/exclusion criterion in literature search and screening, since this disease’s dynamics complexity is a priory expected or known? If you refer to your initial eligibility criteria stated (inclusion of at least one spatially relevant –environmental- factor ...) this might pose some degree of contradiction. Nevertheless, this could be suggested to be applied and serve as such criterion in future relevant research.   

Line 310: I would suggest the use of the term “breeding sites” instead, since breeding sites are included and are part of habitats. 

Line 327: I assume the term “mixed” refers to inconsistent or in different direction effects of specific factors. The measurement of climatic / environmental variables can be more objective and quantitative, especially with the use of standardized instruments, yet demographic and socioeconomic factors may be subject to significant variation due to different case definitions, scoring or categorization of the recorded information which is comparatively more subjective with less distinct threshold limits. 

Line 347: Healthcare access has been mentioned and include in the investigation for identification of potential risk factors, yet for the majority of readers, less familiar with structure of PICTs communities, there is the query of the level and quality or deficiencies of existing healthcare infrastructures and services not only the degree of accessibility to them. Therefore, it would be important to incorporate in the context of this part of the discussion, not only the access that citizens of these regions have to health services (prevention, treatment, information , care etc.), but the level, quality and capacity of actual existing health system and policies, not necessarily limited to BVDs. Considering the aforementioned disproportionality in exposure to adverse effects of climatic change, this could also be another ongoing disproportionality, especially compared to modern Western or in general developed societies. 

Author Response

Reviewer's comment: The submitted manuscript provides a well-designed, properly executed and thoroughly, at least according to set objectives, review; charting up-to-date literature and comprehensibly summarizing relevant existing research-based evidence. Practically, it can be said that its primary purpose as a scoping review has been fulfilled to a large degree. Nevertheless, you will find below some general and specific comments and remarks, aiming to further enhance the validity and quality of your work. 

Authors’ response: We sincerely thank the reviewer for the positive and encouraging assessment of our manuscript. We greatly appreciate the recognition of the study’s design, execution, and its contribution to mapping and synthesizing the current literature. In response to the general and specific comments provided, we have carefully revised the manuscript to further enhance its clarity, methodological rigor, and overall quality. All comments have been addressed in detail below, and the corresponding changes have been highlighted in the revised manuscript. We are grateful for the reviewer’s constructive feedback, which has significantly strengthened our work.

Reviewer's comment: Line 39: For the shake of uniformity and consistency in the use and definition of PICT acronym, “countries” should be written as “Countries”. 

Authors’ response: Thank you for this comment. We confirm that the term “Countries” is now written with an initial capital letter, as follows:

“The Pacific Island Countries and Territories (PICTs) comprise …”

Reviewer's comment: Lines 41-43: I believe a misinterpretation has taken place here. I believe, according to available information, this ratio is incorrect. The landmass area of PICTs is approximately 566.000 km2 within a broader ocean area of 29*106 km2. Actually, the Pacific Ocean’s surface coverage is one-third of the Earth’s surface (165.2 * 106 km2 / 510*106 km2, respectively). Please amend.
Authors’ response: Thank you for this important clarification. To avoid any potential misinterpretation related to surface area estimates, we have revised the text to remove the specific area values and to clearly distinguish between the vast spatial extent of the Pacific Ocean, which covers nearly one-third of the Earth’s surface, and the much smaller landmass of the Pacific Island Countries and Territories (PICTs). The revised text also retains the population estimate of the PICTs.

“The Pacific Island Countries and Territories (PICTs) are a group of small island landmasses dispersed across the vast Pacific Ocean, which covers nearly one-third of the Earth’s surface [1]. Despite their wide geographic distribution, the PICTs are home to approximately 11.4 million people and are predominantly tropical and subtropical in climate [2–4].”

Reviewer's comment: Line 46: Disproportionately in what terms? I agree with your stated opinion, but it remains unclear what this disproportionate “ratio” refers to. Is it in terms of land surface coverage, population density, level of economic development or status, produced environmental impact of their socioeconomic activity etc.? I believe the imposed exposure and impact of climatic change is disproportionate in all of these senses and more, thus a more integrated and global approach will be needed to mitigate this burden. 

Authors’ response: We thank the reviewer for this valuable clarification. We agree that the term “disproportionately” required further specification. The text has now been revised to explicitly clarify that the disproportionate burden refers to a combination of geographic exposure, economic constraints, and limited adaptive capacity. The revised text reflects a more integrated interpretation of climate-related vulnerability in the PICTs.

“PICTs experience a disproportionate burden of climate change impacts due to the combined effects of high geographic exposure (e.g., low-lying coastal settlements), economic constraints, and limited adaptive capacity. These challenges manifest through rising sea levels, extreme weather events, and climate-sensitive livelihoods, and are further compounded by demographic and socioeconomic factors [5].”

Reviewer's comment: Lines 145-146: It is frequent in analytical studies, for most readers to perceive and several researchers to use the terms ‘multi-variate” and “multi-variable” interchangeably or as synonyms erroneously. In fact, these two terms refer to and define different situations; the former concerns statistical analyses procedures or models that encompass the simultaneous observation and analysis of more than one outcome variable (with potentially multiple independent variables), whereas the second term involves one outcome (or dependent) variable and multiple independent (predictor) variables. Judging from the content of Table 1 and 2, through the “Disease” and “Outcome” columns, I assume you refer to studies that investigated more than one VBD and/or measured more than one measure of disease as outcome, thus using the term as suggested above. If this is the case, and since both terms are mentioned in your manuscript (see also line 289), it would be useful to avoid the preservation and continuation of such misconceptions, by briefly explaining or defining the actual use of the term. The use of the term “multivariable” in line 289, seems appropriate and sensible though, in the provided context of the rationale of pose statement. 

Authors’ response: We thank the reviewer for this important methodological clarification regarding the distinction between multivariate and multivariable analyses. We fully agree with the definitions provided. In our study, the analyses reported across the included articles predominantly involved a single outcome variable with multiple predictors, and therefore the correct term is multivariable, not multivariate. To avoid any potential ambiguity or misuse of terminology, we have now revised the manuscript to ensure consistent use of the term “multivariable” throughout. Additionally, we have explicitly clarified this distinction in Section 2.6 (Data synthesis and analysis) by defining bivariate analyses as unadjusted two-variable associations and multivariable analyses as models involving a single outcome with multiple covariates.

The following clarification was added into the text:

“Where available, we also noted whether the associations were derived from bivariate (unadjusted) or multivariable-adjusted analyses (i.e., models with a single outcome and multiple covariates).”

Reviewer's comment: Lines 301-309: You previously reported in the inclusion criteria of the selected and retained literature, the application of either bivariate or multivariate analyses along with the identification or verification of complex and multivariable interactions among the various types of potential risk factors, through synthesized evidence of the revised literature. Yet, some exceptions from consistent effects of these factors were observed. According to your statement (lines 306-309) such sources are implied to have implemented methodology (either in data collection and/or analysis) that cannot account properly or adequately the interaction or confounding effects of relevant factors (i.e. as temperature with other climatic factors. Shouldn't this be considered a priory as an inclusion/exclusion criterion in literature search and screening, since this disease’s dynamics complexity is a priory expected or known? If you refer to your initial eligibility criteria stated (inclusion of at least one spatially relevant –environmental- factor ...) this might pose some degree of contradiction. Nevertheless, this could be suggested to be applied and serve as such criterion in future relevant research.   

Authors’ response: We thank the reviewer for this thoughtful observation regarding the complexity of disease dynamics and the ability of included studies to adequately account for interaction and confounding effects. We fully agree that vector-borne disease transmission is inherently complex and that, ideally, analytical approaches should explicitly address interactions among climatic, environmental, and socioeconomic factors.

However, as clearly stated in our eligibility criteria (Section 2.2), the aim of this scoping review was to map the breadth of available quantitative evidence based on the inclusion of at least one spatially relevant factor, rather than to restrict inclusion to studies employing a specific level of analytical complexity or interaction modelling. As such, requiring the formal assessment of interaction or advanced multivariable structures as an a priori inclusion criterion would have excluded a substantial proportion of the existing literature and limited our ability to identify important methodological gaps, which is a central objective of scoping reviews.

The statement in Lines 306–309 referring to inconsistent effects across studies was therefore intended as a synthesis finding rather than as a criticism of study eligibility, highlighting how limited adjustment for confounding and interaction may contribute to heterogeneous results in the literature. To avoid any perceived contradiction with the inclusion criteria, we have now clarified this interpretation in the revised manuscript in the Discussion as follows:

“Although the transmission dynamics of VBDs are inherently complex and shaped by interactions among climatic, environmental, demographic, and socioeconomic factors, the present study was designed with the primary aim of mapping the breadth of spatially relevant quantitative evidence in the PICTs. Accordingly, our eligibility criteria required only the inclusion of at least one spatially relevant factor and the reporting of quantitative statistical associations. This inclusive approach allowed us to capture studies with varying levels of analytical sophistication, including both bivariate and multivariable analyses. Importantly, the observed heterogeneity and, in some cases, inconsistency in reported effects across studies should not be interpreted as methodological shortcomings warranting exclusion, but rather as a key synthesis finding, reflecting the limited consideration of confounding and interaction effects in parts of the current evidence base. These findings highlight an important methodological gap in the region and underscore the need for future analytical studies and systematic reviews to explicitly model interaction structures and confounding mechanisms to better reflect the known complexity of vector-borne disease systems.”

Reviewer's comment: Line 310: I would suggest the use of the term “breeding sites” instead, since breeding sites are included and are part of habitats. 

Authors’ response: We thank the reviewer for this comment. We confirm that the text has been now revised accordingly as follows:

“Precipitation is also critical due to the creation of breeding sites for vectors, particularly mosquitoes [11].”

Reviewer's comment: Line 327: I assume the term “mixed” refers to inconsistent or in different direction effects of specific factors. The measurement of climatic / environmental variables can be more objective and quantitative, especially with the use of standardized instruments, yet demographic and socioeconomic factors may be subject to significant variation due to different case definitions, scoring or categorization of the recorded information which is comparatively more subjective with less distinct threshold limits. 

Authors’ response: We thank the reviewer for this insightful clarification. We agree that the term “mixed” should be interpreted as referring to inconsistent or directionally different effects reported across studies. We also fully agree that climatic and environmental variables are generally measured more objectively and quantitatively, while demographic and socioeconomic variables are more susceptible to heterogeneity due to variation in case definitions, scoring, and categorisation across studies. We have now revised the text to explicitly clarify this distinction and to better reflect the role of measurement variability in contributing to the observed inconsistency of findings.

“Mixed findings across studies, referring to inconsistent or directionally different effects of specific factors, may partly reflect differences in methodological approaches, including variation in the spatial scale of analysis, for instance at the neighbourhood compared to regional level. In addition, the measurement of climatic and environ-mental variables is often more objective and quantitative, typically derived from standardized instruments, whereas demographic and socioeconomic factors are more prone to heterogeneity due to differences in case definitions, scoring systems, and categorisation across studies. Beyond these methodological considerations, several underlying mechanisms may further explain variability in effects. For example, over-crowding may increase contact rates and facilitate vector–human interaction, while being born in the Pacific may reflect behavioural or cultural patterns that influence exposure, such as time spent outdoors [24]. Genetic predispositions, such as G6PD deficiency reported in 16 villages in Vanuatu, further illustrate how long-term en-demic exposure may shape population-level vulnerability [35]. Together, these in-sights can inform more effective and context-specific public health interventions tailored to the unique characteristics of island populations.”

Reviewer's comment: Line 347: Healthcare access has been mentioned and include in the investigation for identification of potential risk factors, yet for the majority of readers, less familiar with structure of PICTs communities, there is the query of the level and quality or deficiencies of existing healthcare infrastructures and services not only the degree of accessibility to them. Therefore, it would be important to incorporate in the context of this part of the discussion, not only the access that citizens of these regions have to health services (prevention, treatment, information , care etc.), but the level, quality and capacity of actual existing health system and policies, not necessarily limited to BVDs. Considering the aforementioned disproportionality in exposure to adverse effects of climatic change, this could also be another ongoing disproportionality, especially compared to modern Western or in general developed societies. 

Authors’ response: We thank the reviewer for this important contextual clarification. We fully agree that healthcare vulnerability in the Pacific Island Countries and Territories cannot be adequately characterised by physical accessibility alone, and that the capacity, quality, and functional performance of health systems represent additional and critical dimensions of health risk. While our review focused on spatially measurable indicators of access, we acknowledge that broader system-level constraints, including workforce limitations, infrastructure capacity, service continuity, and policy implementation, may substantially modify the effectiveness of prevention and treatment services in the region, particularly under climate stress. We have now expanded the Discussion to explicitly distinguish access from health system capacity and quality, and to highlight this as a key dimension of disproportional health vulnerability and a priority for future spatial and health systems research.

The following part in the discussion has been expanded:

“Despite growing evidence on the environmental and social determinants of VBDs, critical gaps persist in the literature. Chief among these is the limited exploration of healthcare access as a determinant of disease risk [39,40], a dimension that remains strikingly under-explored. Importantly, healthcare access should not be interpreted solely as physical or geographic accessibility [41], but also in terms of the capacity, quality, and functional readiness of health systems, including workforce availability, continuity of services, surveillance infrastructure, and policy implementation [42]. In many PICTs, structural health system constraints may substantially limit not only access to prevention and treatment, but also the effectiveness and timeliness of care, particularly during climate-sensitive outbreaks and extreme weather events [6]. This represents an additional layer of disproportional vulnerability when compared with high-income Western health systems that typically have greater surge capacity and system resilience.

The limited consideration of healthcare access and system capacity is further ex-acerbated by broader methodological weaknesses in the literature, including inconsistencies in how spatial factors are defined and measured, the lack of harmonised statistical approaches, and a pronounced geographic bias, with most studies concentrated in Papua New Guinea, New Caledonia, and American Samoa. Collectively, these shortcomings constrain the comparability of findings and obscure the regional picture. Future research must therefore move beyond these limitations by integrating healthcare access alongside health system capacity and quality as central determinants, adopting standardised spatial and statistical methods, and broadening geographic coverage to generate a more comprehensive and equitable evidence base.”

Round 2

Reviewer 1 Report

Comments and Suggestions for Authors

The authors thoroughly responded to every comment, making the necessary adjustments across the manuscript. As a result, these changes enhanced the study's clarity, strengthened its methodological approach, and improved how the research is presented overall.

Reviewer 2 Report

Comments and Suggestions for Authors

Dear authors,

Thank you for the thorough response to my posed comments and suggestions. Most importantly, I am grateful for the acknowledgment and acceptance of the rationale of any criticism that was perceived as was actually intended; to improve the quality of your work.